# Probability Characteristics, Area Reduction, and Wind Directionality Effects of Extreme Pressure Coefficients of High-Rise Buildings

**Shouke Li [1,2,\*], Feipeng Xiao [1], Yunfeng Zou [3], Shouying Li [2], Shucheng Yang [4], Chao Feng [4] and Yuankun Chen [5]**

1 School of Civil Engineering, Hunan University of Science and Technology, Xiangtan 411201, China; XFP729624418@hotmail.com
2 College of Civil Engineering, Hunan University, Changsha 410082, China; shyli@hnu.edu.cn
3 School of Civil Engineering, Central South University, Changsha 410075, China; yunfengzou@csu.edu.cn
4 Highway College, Chang'an University, Xi'an 710064, China; syang325@chd.edu.cn (S.Y.); cfeng@chd.edu.cn (C.F.)
5 Central-South Architectural Design Institute Co., Ltd., Wuhan 430071, China; hustkun1012@hotmail.com
\* Correspondence: lishouke@hnust.edu.cn

**Abstract:** Wind tunnel tests are carried out for the Commonwealth Advisory Aeronautical Research Council (CAARC) high-rise building with a scale of 1:400 in exposure categories D. The distribution law of extreme pressure coefficients under different conditions is studied. Probability distribution fitting is performed on the measured area-averaged extreme pressure coefficients. The general extreme value (GEV) distribution is preferred for probability distribution fitting of extreme pressure coefficients. From the comparison between the area-averaged coefficients and the value from GB50009-2012, it is indicated that the wind load coefficients from GB50009-2012 may be non-conservative for the CAARC building. The area reduction effect on the extreme wind pressure is smaller than that on the mean wind pressure from the code. The recommended formula of the area reduction factor for the extreme pressure coefficient is proposed in this study. It is found that the mean and the coefficient of variation (COV) for the directionality factors are 0.85 and 0.04, respectively, when the orientation of the building is given. If the uniform distribution is given for the building's orientation, the mean value of the directionality factors is 0.88, which is close to the directionality factor of 0.90 given in the Chinese specifications.

**Keywords:** wind effects; high-rise building; extreme pressure coefficients; probability distribution; area reduction factor; wind directionality factor

## 1. Introduction

The wind effects on high-rise buildings are widely studied by many researchers [1–4]. The study of extreme pressure coefficients is a focus of extreme wind effects for high-rise buildings. The extreme pressure coefficient for a high-rise building due to extreme local wind pressures on its surface is uncertain. The quantification of the uncertainty, i.e., the probabilistic characterization, for such a random variable is important for calibrating the design wind load for the codes. This usually includes the probabilistic distribution fitting practice for the extreme pressure coefficients obtained from numerous wind tunnel tests on the buildings or from the literature. For example, Cook and Mayne [5,6] attempted to fit the extreme pressure coefficient for low-rise buildings by using the Gumbel distribution. To estimate the distribution parameters, they applied the best linear unbiased estimation (BLUE) method proposed by Peterka [7]. It is an optimized method for prediction of peak pressure from wind tunnel model tests. Similar research has been performed by Holmes and Cochran [8] considering the generalized extreme value (GEV) distribution with positive shape parameters. Chen and Huang [9] studied the effects of different probability distributions of wind speed and extreme load coefficients on the extreme wind

effects. This indicated that the use of the extreme Type III distribution for the extreme load coefficient had almost no influence on the wind effects. Quan et al. [10] suggested that the probability distribution of the extreme value of wind pressure on the surface of low-rise buildings could be expressed using Gumbel distribution and GEV distribution. Wu [11] found that the probability distribution of extreme wind pressure on long-span roofs could be expressed using the GEV distribution. The GEV distribution gave the best fit for the extreme peak data of pressure for wind-induced acceleration analysis [12]. However, rare examples can be found for the probabilistic characterization of the extreme pressure coefficients for high-rise buildings from the literature.

To estimate the extreme wind pressures on the claddings of high-rise buildings from the wind tunnel test, one of the issues is that the measurements are usually different between point pressures and spatial averages. Due to the lack of full spatial and temporal correlations, the localized intense pressures (e.g., point pressures) are likely to be attenuated for multiple surfaces (e.g., spatial averages) [13]. Such an attenuation effect is captured by several building design codes. An example can be found in ASCE 7–10 [14] in the provisions for the design of components and cladding, indicating that the pressures decrease exponentially with area for the design pressure coefficients. To accurately account for the attenuation effect, Lawson [15] and Holmes [16] concluded that the equivalent time averaging method is good for computing wind loads on finite areas of structures. However, Kopp and Morrison [13] indicated that area-averaged pressure coefficients can be used to determine cladding and component loads. The study shows that as the subordinate area of the component increases, the total wind load effect will decrease. "Load Code for the design of building Structure" GB50009-2012 [17] gives the area reduction factor of the surface local shape factor for taking into account the area reduction effects for wind load effects. The local shape factor is the mean pressure coefficient with the reference height of its own. As for the area reduction of the extreme pressure coefficient, GB50009-2012 lacks a corresponding quantitative description of the reduction relationship.

In addition to the attenuation effect of the varying area on the surface pressures of the buildings, pressures may also differ significantly as a function of the wind direction. To account for such effects, design guidelines conservatively use an enveloping approach, in which the extreme pressure coefficients over all wind directions are assessed and recommended for estimating the wind pressures. A typical value of 0.85 is suggested by ASCE 7–10 [14], which is agreed on by Isyumov [18], for high-rise buildings considering synoptic wind pressures. Zhang and Chen [19] proposed a new approach of estimating wind load effects with a consideration of the directionality of wind. It is acknowledged that the wind directionality effect on the wind load should be considered for structural design. The 0.90 of wind directionality factor is included in GB50009 (2012). Is it suitable for the extreme wind pressure of the high-rise building envelope structure, which needs further study.

To systematically evaluate the extreme pressures on high-rise buildings, a schematic wind tunnel experimental study is proposed and conducted. By adopting the test results to probabilistically characterize the extreme pressure coefficients, the objective of this study is to calibrate the design wind loads, considering the effect that the pressures decrease with area, as well as the effect of wind direction. A directionality factor is also recommended for the current code of China.

The layout of this paper is as follows: Section 2 introduces the experiment setup and data analysis; Section 3 introduces the results and discussion about probability characteristics, area reduction effects, and wind directionality effects; and Section 4 consists of concluding remarks.

## 2. Experiment Setup and Data Analysis

### 2.1. Wind Tunnel Test Setup

The schematic wind tunnel experimental study program was conducted in the Building Aerodynamics Laboratory of Hunan University of Science and Technology. Considering

the simulated flow properties for the specific wind tunnel, a geometric scale of 1:400 is selected, and the wind tunnel tests are performed for the most critical suburb terrain exposure D in load code for the design of building structures (GB50009-2012). The atmospheric boundary layer was modelled at a geometric scale of 1:400. Some devices (spires, vortex generators, and roughness elements) were placed at the entrance to the test section to generate acceptable mean and turbulent flow condition [20]. The mean velocity and turbulence intensity profiles matched the specification of GB50009-2012 with power law of $\alpha = 0.30$ and $I_0 = 0.39$ for exposure D, which are shown in Figure 1a. $\alpha$ is the power law index. $I_0$ is the turbulence intensity at the height of 10 m, and H is the height of building. Figure 1b shows the spectrum of the longitudinal wind velocity fluctuations, compared with the Karman spectrum.

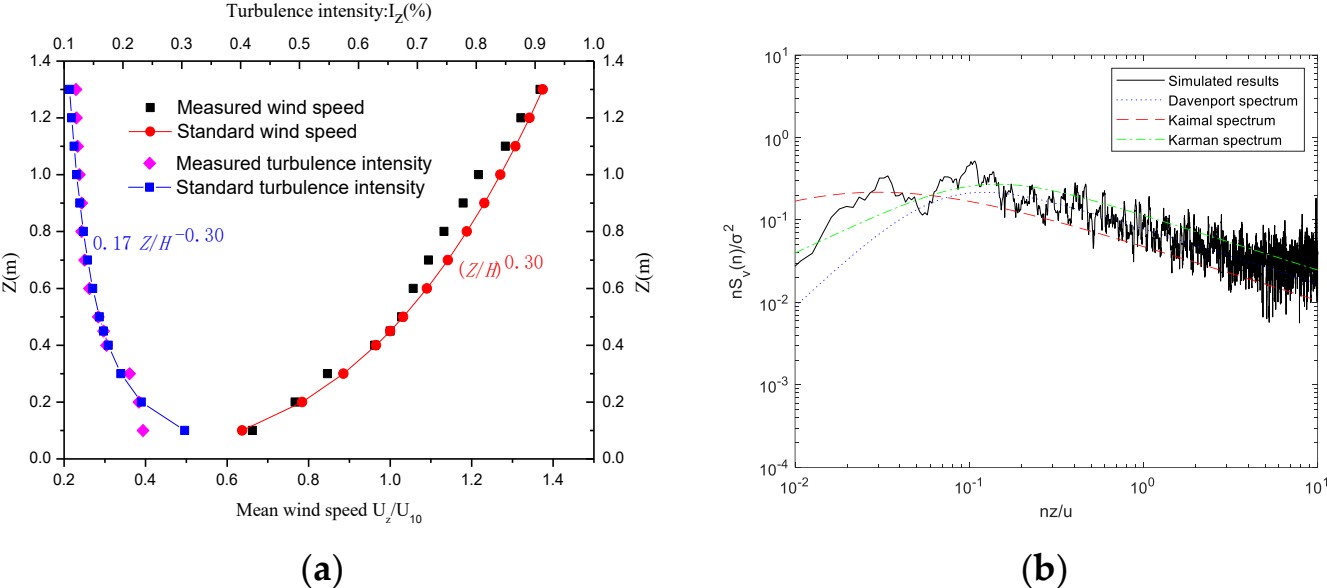

**Figure 1.** Approach flow condition for exposure D. (**a**) Mean velocity and turbulence intensity profiles; (**b**) Longitudinal wind velocity fluctuations spectrum at 2/3 building height.

The full-scale dimensions of the modelled Commonwealth Advisory Aeronautical Research Council (CAARC) high-rise building are 30.48 m × 45.72 m × 182.88 m (length × width × height). The scale ratio for the wind tunnel test model is 1:400. The size in model scale was 76.20 mm × 114.30 mm × 457.20 mm (length × width × height). The dimensions of test section for the wind tunnel are 4 m width × 3 m height, resulting in the blockage ratio of wind tunnel tests being 0.29~0.44%. Figure 2 shows the experimental model for the wind tunnel test. The model building is equipped with a total of 308 pressure taps, and the swarmed pressure taps were arranged at 2/3 the height of the building. Figure 3 shows the building sizes, the positions of the pressure measuring taps, the number of the test points in the swarmed area, and the wind direction arrangement. In the corner diagram, the four facades of the building are named East E, West W, South S, and North N. The wind direction angle is 0° when facing northward.

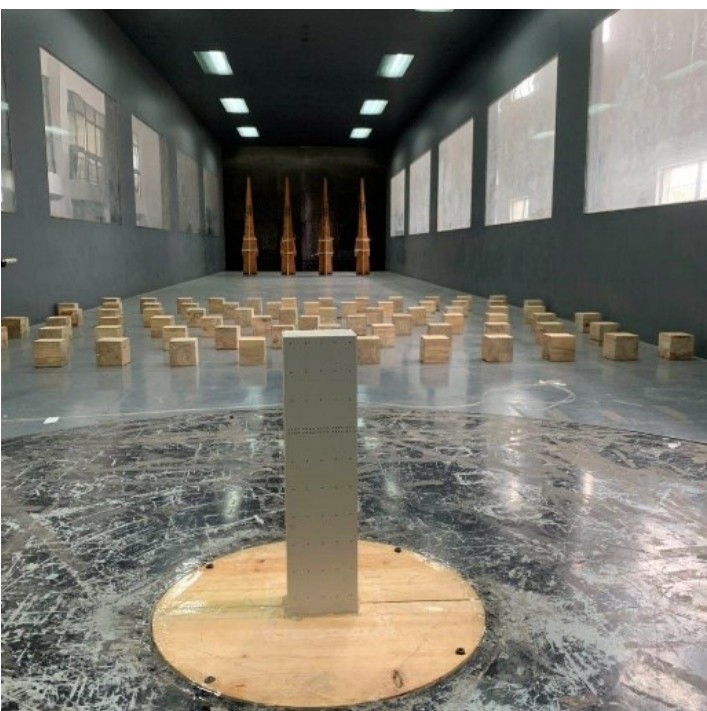

**Figure 2.** Experimental model.

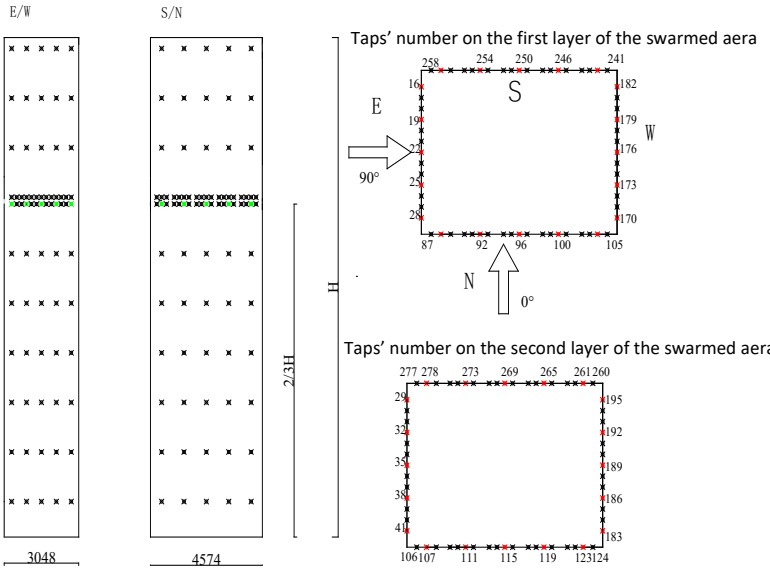

**Figure 3.** Measuring tap location for scale model (unit: cm).

The sampling frequency of the test is 330 Hz, and each measurement tap is sampled for about 60 s. Sixty seconds for wind tunnel tests is equivalent to one hour in full scale. The total length of one sample is 20,000 data points, and each wind direction angle is independently sampled 20 times repeatedly. In the tests, the reference height is selected at 45.7 cm to the bottom of the test model, which is the height of the building.

### 2.2. Data Analysis

The time series of the pressure coefficients $C_{pi}(t)$ are defined as follows [21]:

$$C_{pi}(t) = \frac{p_i(t) - p_0}{0.5 \rho u_h^2} \tag{1}$$

where $p_i(t)$ is the time series of the wind pressure measured by the pressure scanning valve in the wind tunnel test; $p_0$ is the mean value of static pressure at the wind tunnel test section, which is measured using a pitot tube; $\rho$ is the air density, taken $\rho = 1.225\text{kg/m}^3$; and $u_h$ is the mean wind speed at the reference height. The reference height is the height of building. The extreme pressure coefficients of taps are defined as follows:

$$\text{C}_{\text{pmax}} = \frac{p_{\max} - p_0}{0.5\rho u_h^2} \tag{2}$$

$$\text{C}_{\text{pmin}} = \frac{p_{\min} - p_0}{0.5\rho u_h^2} \tag{3}$$

The time series of the area average wind pressure coefficient are defined as follows:

$$C_f(t) = \sum_{i=1}^{n} [C_{P,i}(t) \times A_i] / \sum_{i=1}^{n} (A_i) \tag{4}$$

where $A_i$ is the tributary area of the tap number $i$, and $i = 1, 2, 3, \ldots n$ ($n$ represents total number of taps).

The extreme pressure coefficients of taps or area-average are calculated by the extreme value probability distribution fitting method using 20 independent sampling peaks, where the distribution parameter is determined by the maximum likelihood method. The result is taken as 78% of the extreme quantile.

## 3. Results and Discussion

### 3.1. Probability Characteristics of Extreme Pressure Coefficients

#### 3.1.1. Extreme Value Distribution Type

Three extreme value probability models are used to determine the probability distribution of the sample extreme value of wind load coefficient and the quantiles, which are Extreme value type I distribution (Gumbel distribution), Extreme value type II distribution (Frechet distribution), and Extreme value type III distribution (Weibull distribution). In this study, the model is described as Gumbel distribution when the shape parameter is 0, and as the GEV distribution when the shape parameter is not 0. The specific expressions of the three extreme value distributions are shown in Table 1.

**Table 1.** Extreme value probability models.

| Type of Distribution | Form | Condition |
|---|---|---|
| Extreme value Type I distribution: Gumbel distribution | $F(x) = \exp\{-\exp[-(x-u)/a]\}$ | $x \in R$ |
| Extreme value Type II distribution: Frechet distribution | $F(x) = 0$ <br> $F(x) = \exp\left[-(x-u)/a\right]^{-1/k}$ | $x \leq u$ <br> $x > u, k > 0$ |
| Extreme value Type III distribution: Weibull distribution | $F(x) = \exp\left[-(x-u)/a\right]^{-1/k}$ <br> $F(x) = 0$ | $x < u, k < 0$ <br> $x \geq u$ |

Note: The three extreme value distributions can be expressed in a unified form GEV distribution: $F(x) = \exp\left\{-[1+k(x-u)/a]^{-1/k}\right\}$, where $u$ is the position parameter; $a$ is the scale parameter; and $k$ is the shape parameter. When $k = 0$: extreme value Type I distribution; when $k > 0$: extreme value Type II distribution; when $k < 0$: extreme value Type III distribution.

#### 3.1.2. Probability Distribution Modelling of Extreme Pressure Coefficient

(1) The probability distribution model of the area-averaged extreme pressure coefficient for different areas for a typical wind angle.

Probability distribution fitting is conducted for the extreme area-averaged wind pressure coefficients in a swarmed area. Figure 4 shows the probability distributions of the area average extreme pressure coefficients of 2 m², 10 m², 25 m², and 72 m² (full scale) in the swarmed area of the west facade at 0° wind direction angle. It can be seen from Figure 4

that the area-averaged extreme pressure coefficients for different areas in the swarmed area have good fitting performance using the GEV distribution, and the *k* is less than zero, which is best fitted with the extreme value Type III distribution. As for Gumbel distribution, there is an obvious deviation in some areas.

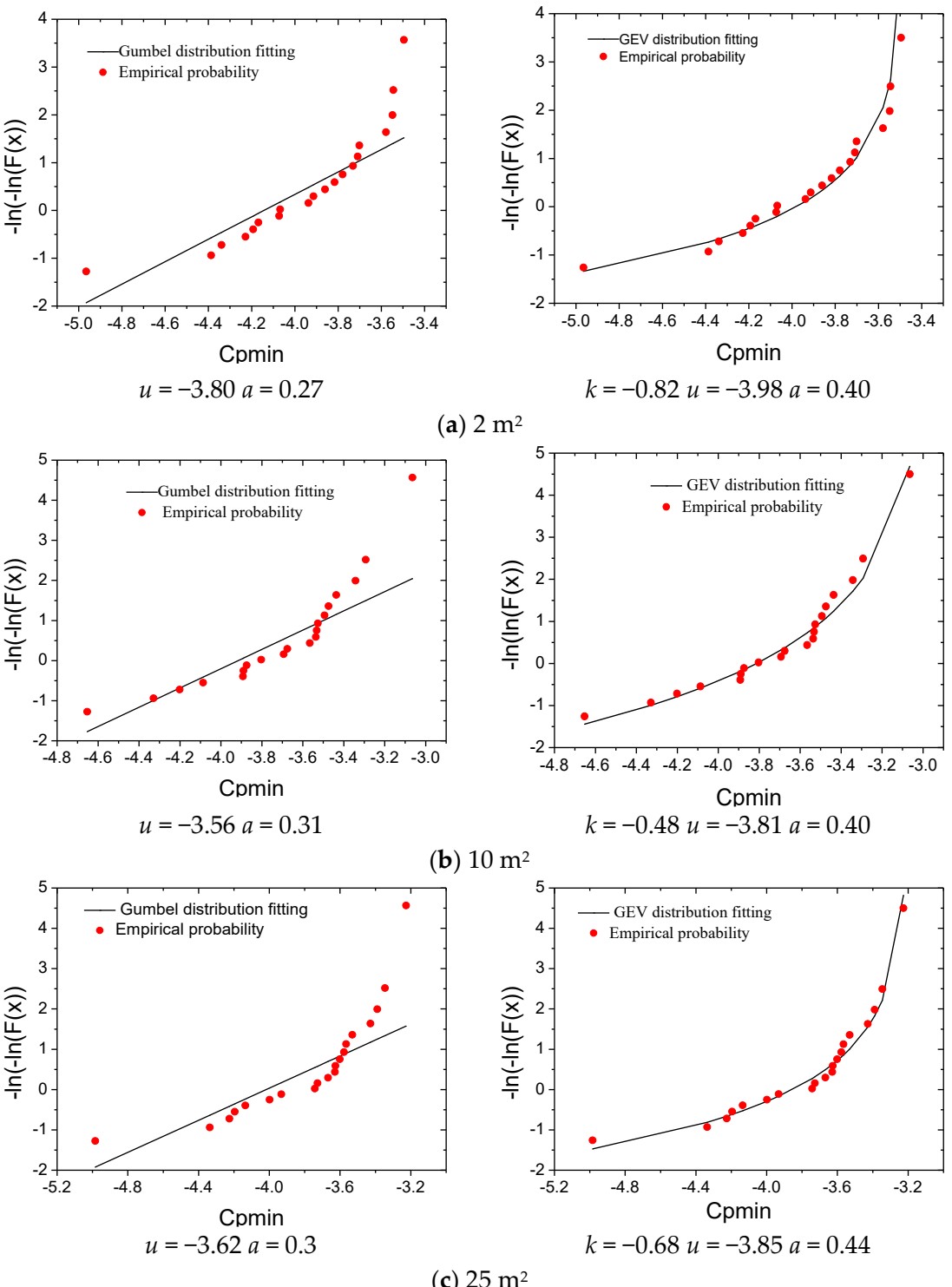

(**a**) 2 m²

(**b**) 10 m²

(**c**) 25 m²

**Figure 4.** *Cont.*

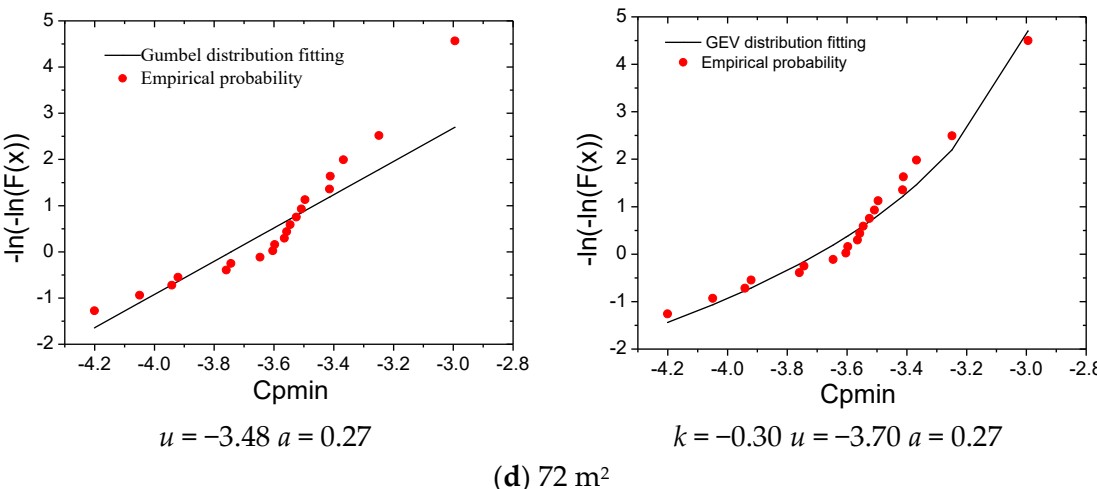

**Figure 4.** The probability distribution fitting for the area-averaged extreme pressure coefficients for different areas: (**a**) 2 m$^2$, (**b**) 10 m$^2$, (**c**) 25 m$^2$, (**d**) 72 m$^2$.

The Kolmogorov–Smirnov (K-S) test is applied to see whether the Gumbel or GEV distribution is preferred. The K-S test expression for the two attempts is as follows:

$$D_K = \max|Fe(x) - F(x)| \tag{5}$$

where *Fe* (*x*) is the hypothetical probability distribution, *F* (*x*) is the cumulative probability distribution of the sample itself, and the standard evaluation formula $D_K < D_K\alpha$. The samples obtained by the wind tunnel test are considered to follow this probability distribution. $\alpha$ is a significant level, and $D_K\alpha$ is above the critical value.

The Akaike information criterion (AIC) can also be used to test the fitting goodness of the extreme value probability distribution [22]. The AIC method is expressed as

$$\text{AIC} = 2n - 2\ln L \tag{6}$$

where *n* represents the number of free parameters of the considered model, and L represents the maximum likelihood of the model.

Figure 5 shows the results of the K-S test method and the AIC test method for the probability distribution fitting of the minimum wind pressure coefficients in the swarmed area for different areas. The results indicate that the use of the GEV distribution is preferred for the probability distribution fitting of the extreme pressure coefficient for all kinds of area sizes.

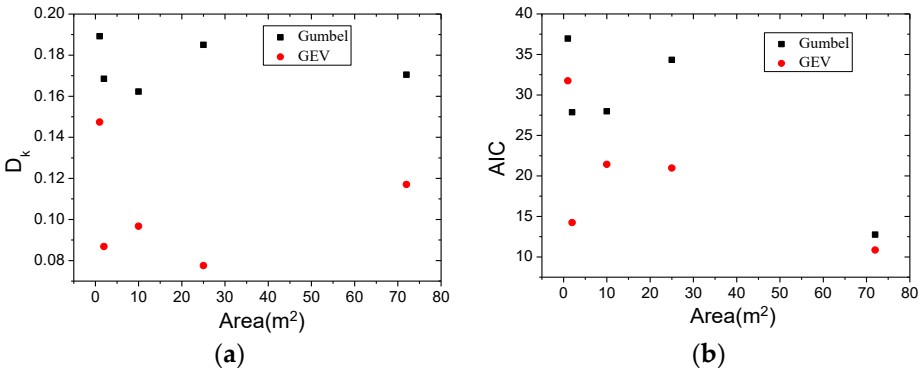

**Figure 5.** Examination of probability distribution fitting goodness of extreme pressure coefficients. (**a**) $D_K$ value in swarmed area. (**b**) AIC value in swarmed area.

(2) Effect of wind direction on the probability distribution model for peak pressure coefficients.

Figure 6 shows the fitting for the Gumbel and GEV distribution with minimum wind pressure coefficients of 2 m$^2$ area in the north facade of the building at several typical wind direction angles (45°, 60°, 90°, 120°, 135°, 180°). As can be seen from Figure 6, the fitting departs considerably from the fitted line for the case of the Gumbel distribution. The fitting is reasonable in the case of the GEV distribution for all wind directions, and the shape parameter value K in the GEV parameter estimation is less than 0.

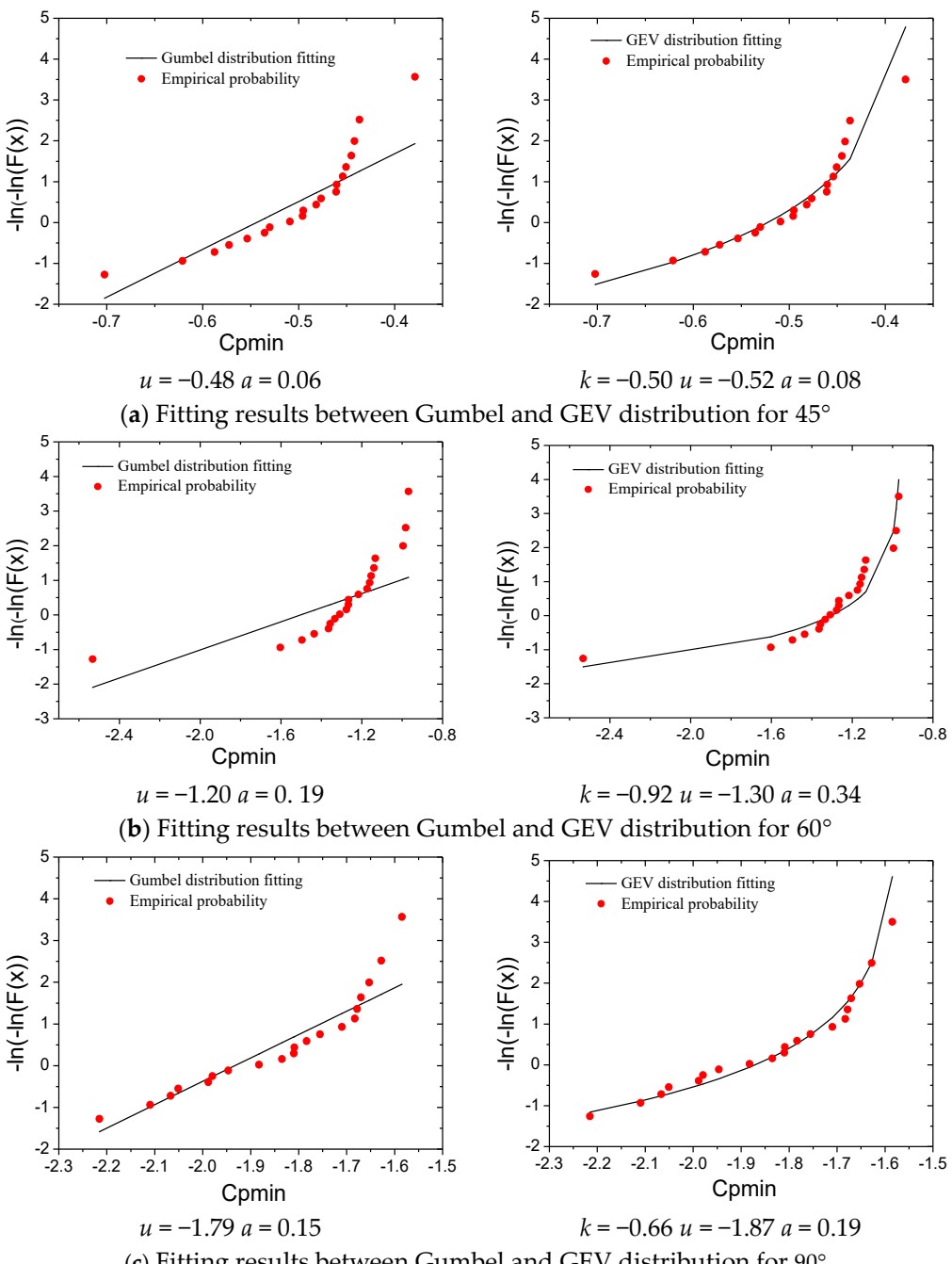

$u = -0.48\ a = 0.06$       $k = -0.50\ u = -0.52\ a = 0.08$

(**a**) Fitting results between Gumbel and GEV distribution for 45°

$u = -1.20\ a = 0.19$       $k = -0.92\ u = -1.30\ a = 0.34$

(**b**) Fitting results between Gumbel and GEV distribution for 60°

$u = -1.79\ a = 0.15$       $k = -0.66\ u = -1.87\ a = 0.19$

(**c**) Fitting results between Gumbel and GEV distribution for 90°

**Figure 6.** *Cont.*

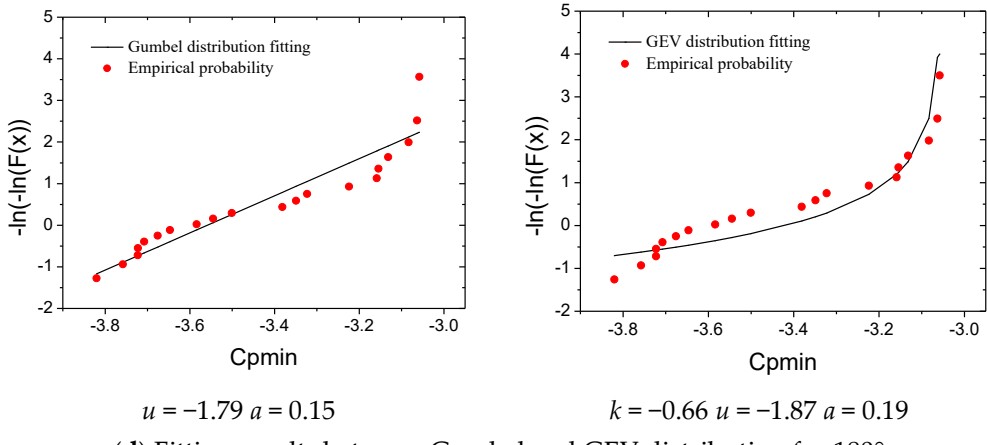

$u = -1.79\ a = 0.15$       $k = -0.66\ u = -1.87\ a = 0.19$

(**d**) Fitting results between Gumbel and GEV distribution for 180°

**Figure 6.** Fitting results of the probability distribution of extreme pressure coefficient under different wind direction angles: (**a**) Fitting results between Gumbel and GEV distribution for 45°, (**b**) Fitting results between Gumbel and GEV distribution for 60°, (**c**) Fitting results between Gumbel and GEV distribution for 90°, (**d**) Fitting results between Gumbel and GEV distribution for 180°.

The K-S test and the AIC method are used to investigate the fitting goodness for the extreme pressure coefficient. Figure 7 shows the probability distribution fitting test chart of the extreme pressure coefficients for different wind directions. It can be seen from Figure 7 that for the wind angle of 60°, the $D_K$ of the Gumbel distribution fitting is slightly larger than that of the GEV distribution, but the difference is small. Combined with the probability plot, it can be seen that the performance for the fitting based on the GEV distribution is better than that based on the Gumbel distribution, and for the wind direction of 90°, the $D_K$ of the Gumbel distribution fitting is significantly greater than that of the extreme value Type III distribution. The fitting goodness for the extreme value Type III distribution is significantly better than that for the Gumbel distribution. There are the same conclusions for the other wind directions. The AIC test can obtain the same conclusions as the K-S test.

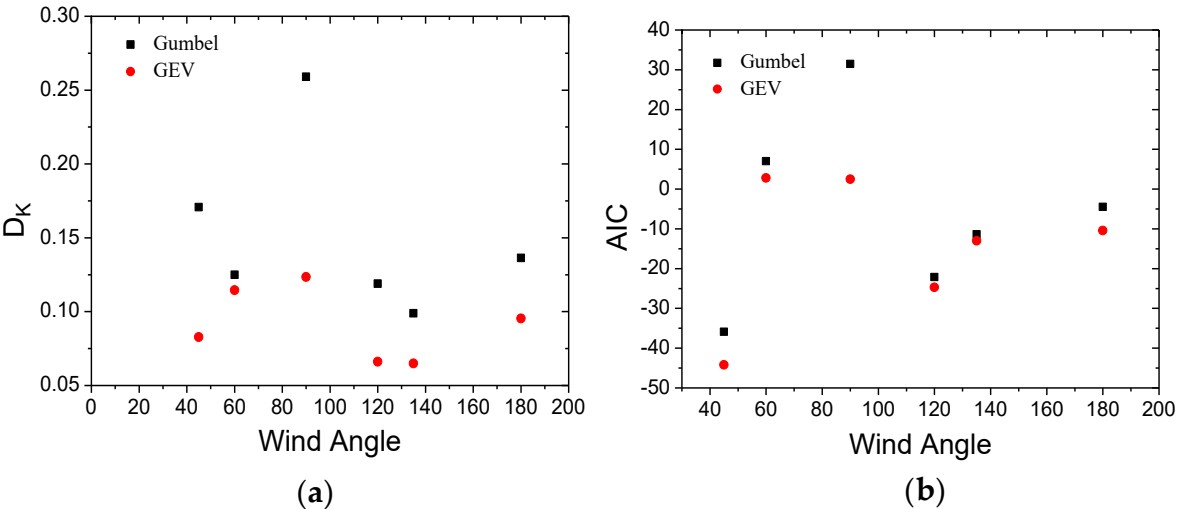

(**a**)            (**b**)

**Figure 7.** $D_K$ value and AIC value of extreme pressure coefficients for different wind angles. (**a**) $D_K$ of K-S test; (**b**) AIC of AIC method.

Therefore, in general, it can be concluded that the wind direction has little effect on the distribution model for peak pressure coefficients. The extreme value Type III distribution is preferred for the extreme pressure coefficient.

(3) Attribution of the probability distribution of the extreme wind pressure on the building envelope.

The case of a wind angle of 0° is selected to study the probability model of the extreme pressure coefficient for an area of 2 m² on the building envelope. Figure 8 shows the ratio of $D_K$ that is based on the Gumbel distribution to the GEV distribution for the building envelope. Table 2 provides the fitting distribution percent of the extreme pressure coefficients on the building envelope.

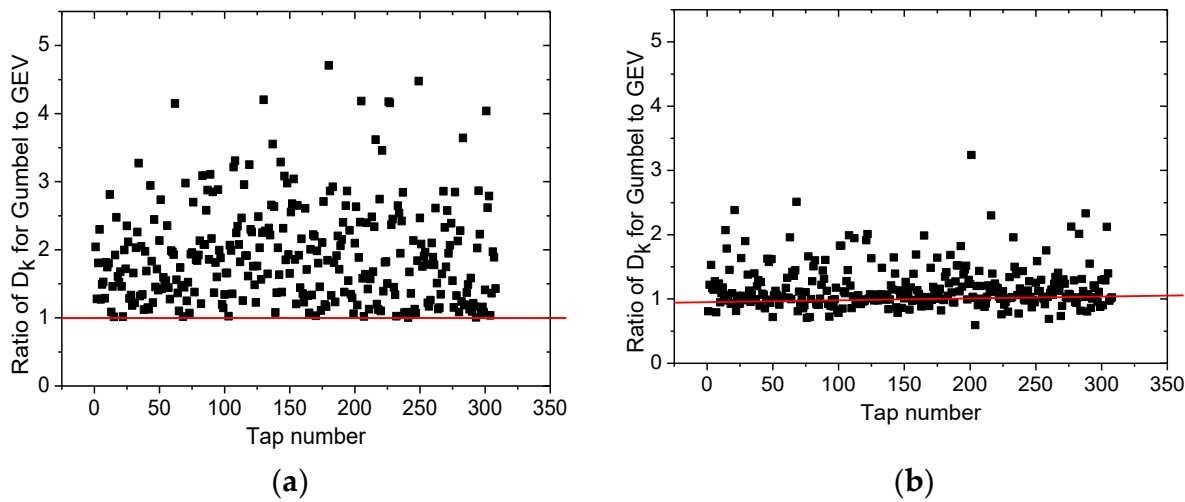

**Figure 8.** Ratio of $D_K$ the Gumbel distribution to the $D_K$ of GEV distribution. (**a**) Minimum pressure coefficient; (**b**) Maximum pressure coefficient.

**Table 2.** Proportion of each probability distribution.

| Extreme Probability Distribution | Extreme Value Type I Distribution | Extreme Value Type II Distribution | Extreme Value Type III Distribution |
|---|---|---|---|
| Minimum | 0 | 0 | 100% |
| Maximum | 7% | 55% | 38% |

It can be observed from Figure 8 that the $D_K$ value obtained by the Gumbel distribution is larger than that based on the GEV distribution. The GEV fitting goodness will be better, and all the minimum wind pressure coefficients belong to the GEV distribution and follow the extreme value Type III distribution. For the maximum pressure coefficient, it is affected by the separation of airflow. The percent of probability distribution for the maximum value is 7%, 55%, and 38% by considering Type I, II, and III distributions consequently.

### 3.1.3. Comparison of Extreme Pressure Coefficients with Current Specifications

For the claddings of high-rise buildings, Chinese code GB50009-2012 recommends using $\beta_{gz}\mu_{sl}$ to represent the extreme pressure coefficient in the wind tunnel test, where $\mu_{sl}$ is the local mean pressure coefficient, and $\beta_{gz}$ is the gust factor at the height $z$.

$$\beta_{gz} = 1 + 2gI_{10}(Z/10)^{-\alpha} \tag{7}$$

where $g$ is the peak factor that is equal to the 2.5 recommended in the GB50009-2012; $I_{10}$ refers to the nominal turbulence intensity at a height of 10 m, taken as 0.39; and $\alpha$ is the index of the power law, taken as 0.30. Taking the 0° wind direction and 90° wind direction as examples, the contour of maximum or minimum extreme pressure coefficients for the windward wall or leeward wall is shown in Figures 9 and 10. The extreme pressure coefficients are compared between the envelope coefficients (EC) from the point pressure coefficients among all wind directions. The area-averaged coefficients (AAC) account for

the area reduction effects of pressure coefficients among all wind directions, as well as the value from GB50009-2012 (GB). The envelope value (EC), the area-averaged value (AAC), and the value from the code (GB) are also shown in Figures 9 and 10.

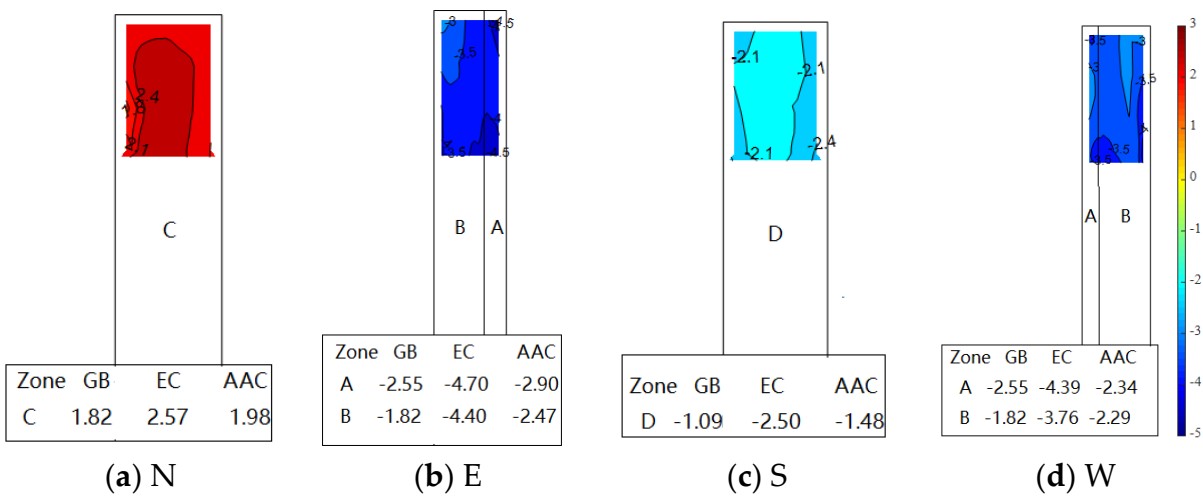

**Figure 9.** Contour of extreme pressure coefficient at 0 ° wind direction angle: (**a**) N, (**b**) E, (**c**) S, (**d**) W.

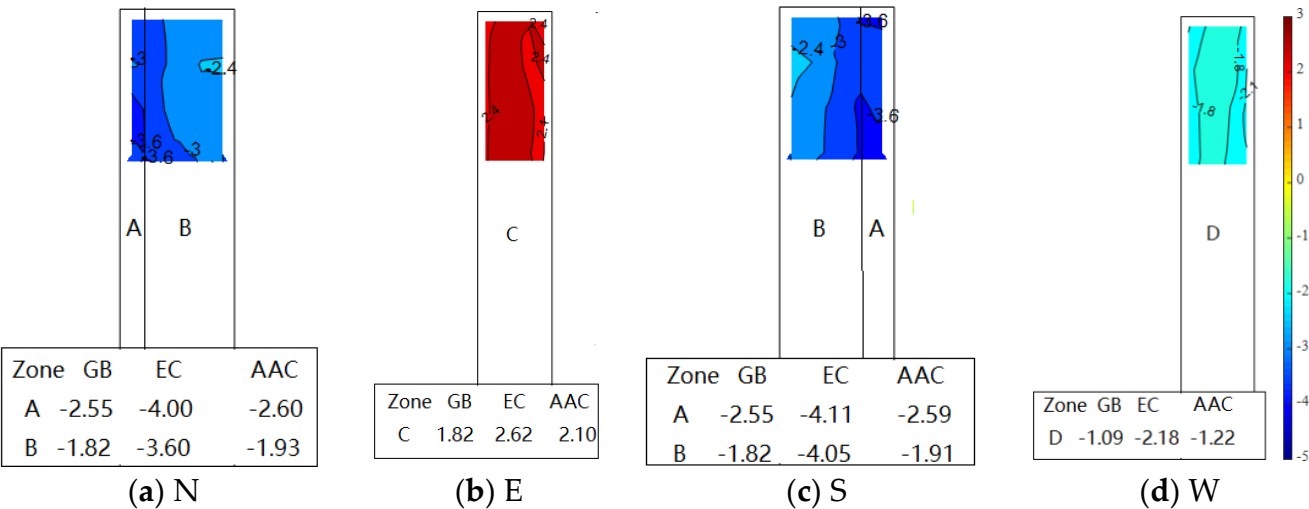

**Figure 10.** Contour of extreme pressure coefficient at 90 ° wind direction angle: (**a**) N, (**b**) E, (**c**) S, (**d**) W.

Figures 9 and 10 show that the area-averaged coefficients (AAC) are smaller than the envelope coefficients. It is indicated that the envelope coefficients are conservative, and the Chinese code may be non-conservative from the comparison between the area-averaged value (AAC) and the value from the code (GB). Furthermore, the Chinese code zoning for the CAARC building facades parallel with the wind direction does not follow the extreme suction distributions measured. There is a need for updating the Chinese code values and zoning, especially for buildings with heights of about more than five times the crosswind dimension similar to the CAARC building.

### 3.2. Area Reduction of Extreme Pressure Coefficient

3.2.1. Extreme Pressure Coefficient for Different Area Sizes under Different Wind Angles

This paper studies the area reduction effects of wind pressure for four faces; the subordinate area of each facade is 1 m$^2$, 5 m$^2$, 10 m$^2$, 25 m$^2$, 33 m$^2$, and 72 m$^2$ (full scale). The different subordinate areas contain several different measuring points. Figure 11 shows the comparison of the area-averaged extreme pressure coefficient under four typical wind

direction angles (0°, 90°, 180°, 270°) for the different facades with different area sizes. It can be seen from Figure 11 that the area-averaged maximum wind pressure coefficient decreases as the area size increases, and the wind pressure on this area decreases. The area-averaged minimum wind pressure coefficient increases as the area size increases, and the suction on this area decreases.

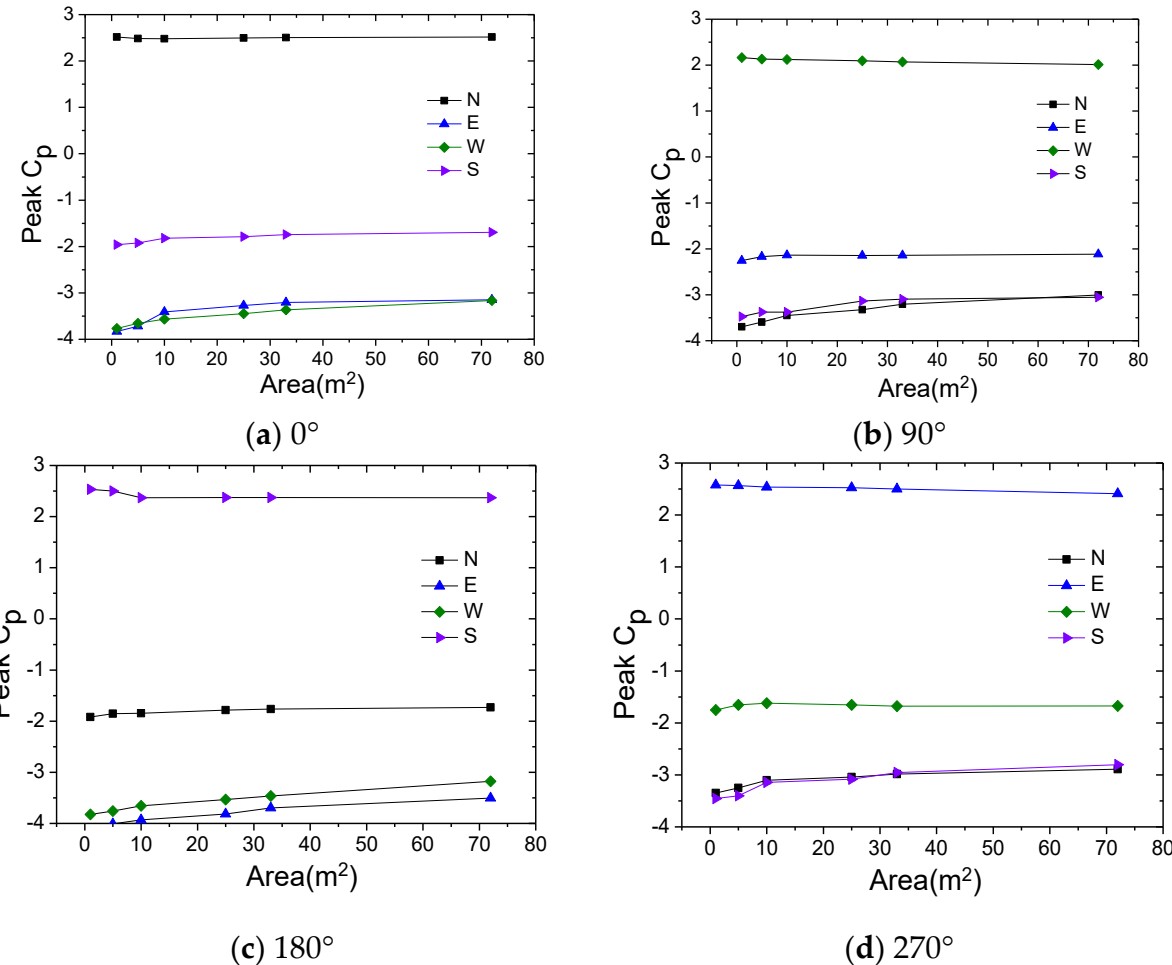

**Figure 11.** Comparison of the area-average peak pressure coefficients of each facade for different area size under different wind direction: (**a**) 0°, (**b**) 90°, (**c**) 180°, (**d**) 270°.

### 3.2.2. Area Reduction Factor of Extreme Pressure Coefficient

To consider the reduction effect of larger areas on the mean wind pressure coefficient, GB50009 (2012) stipulates that the reduction of the design wind load should be considered through the area reduction method of the local shape coefficient. The local shape coefficient refers to the mean wind pressure coefficient and is given by

$$\mu_{sl}(A) = \mu_{sl}(1) - [\mu_{sl}(1) - \mu_{sl}(25)] * \lg A / 1.4 \tag{8}$$

where $\mu_{sl}(A)$ and $\mu_{sl}(1)$ are the local shape coefficients of the area with $A$ m$^2$ and not greater than 1 m$^2$ in the cladding, and the area size of $A$ is less than 25 m$^2$.

From Section 3.2.1, it is obvious that there is an area reduction effect on the area-averaged extreme wind pressure. The diameter of the pressure measuring tap provided on the surface of the building model is 1 to 2 mm, and it is generally considered that the measured wind pressure is equivalent to "point wind pressure". If converted to the full-scale building, the equivalent area of the pressure tap is usually less than 1 m$^2$. The minimum area is specified as 1 m$^2$ in the Chinese code. For cladding with a tributary area

greater than 1 m$^2$, the wind resistance design is carried out according to the maximum point wind pressure on the surface of the cladding, which is conservative. The ratio of the extreme pressure coefficient of any area size to that of 1 m$^2$ is the area reduction coefficient $K_a$ of the wind load.

Figure 12 shows the area reduction factors of the extreme pressure coefficients based on wind tunnel tests and local shape coefficients from the code of the N, E, W, and S directions under different area sizes at 0°, 90°, 180°, and 360° wind angles, respectively.

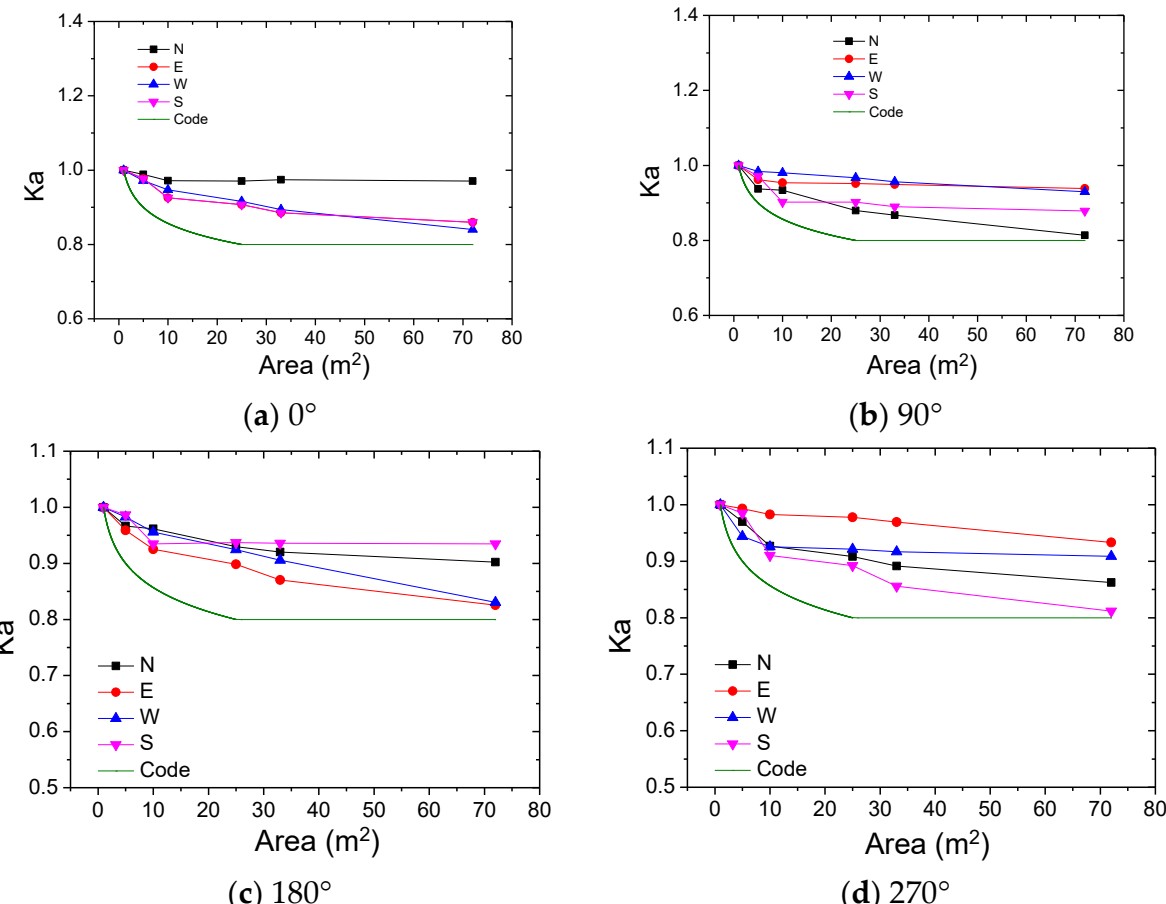

**Figure 12.** The area reduction factors of different area sizes of each facade: (**a**) 0°, (**b**) 90°, (**c**) 180°, (**d**) 270°.

The area reduction factors for the extreme pressure coefficient gradually decrease as the average-areas increase. The results from the comparison show that the reduction effect of the extreme wind pressure is smaller than the area reduction effect of the mean wind pressure. According to GB50009, the extreme wind effect will be underestimated by considering the area reduction of the design wind load of the cladding.

Figure 13 shows the scatter diagram of the area reduction factor of the area-averaged extreme pressure coefficients with areas of 5 m$^2$,10 m$^2$, and 25 m$^2$ under different wind direction angles. As can be seen from Figure 13, when the subordinate area is 5 m$^2$,10 m$^2$, and 25 m$^2$, the extreme wind pressure area reduction factor is in the range of 0.93–0.99, 0.88–0.98, and 0.85–0.97; the mean value of the area reduction factor is 0.97, 0.95, and 0.93; and the COV is 0.02, 0.03, and 0.04. The normal probability distribution is used to fit the area reduction factor $K_a$ of the extreme pressure coefficient. Figure 14 shows the probability plot of the normal distribution of the $K_a$ value. The results shown in Figure 14 indicate that the normal distribution is suggested for the probability distribution fitting of the area reduction factor effect.

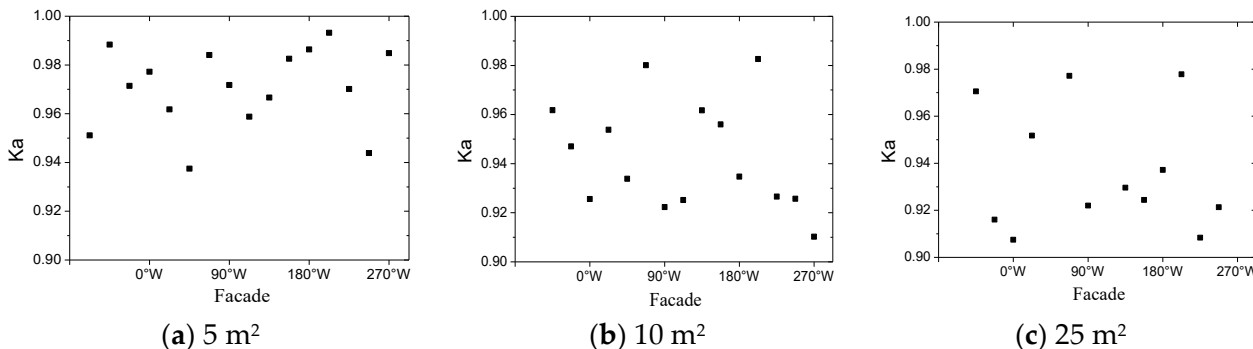

**Figure 13.** Reduction factors of block extremums of different areas under different wind directions: (**a**) 5 m$^2$, (**b**) 10 m$^2$, (**c**) 25 m$^2$.

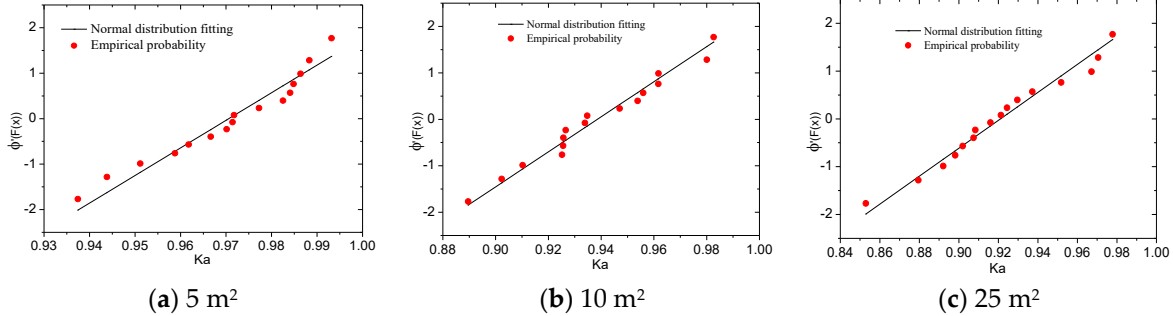

**Figure 14.** Fitting result of the probability distribution of K$_a$ value in different areas: (**a**) 5 m$^2$, (**b**) 10 m$^2$, (**c**) 25 m$^2$.

By transforming Equation (8), the recommended formula of the area reduction factor for the extreme pressure coefficient can be

$$\text{K}_a(A) = 1 - 0.07 * \lg A / 1.4 \tag{9}$$

In the formula, *A* is the area size but is less than 25 m$^2$; K$_a$ takes 0.93 if the area size is larger than 25 m$^2$.

### 3.3. Wind Directionality Effects of Extreme Wind Loads

A directionality factor is often used in building codes (ASCE7-2010) to consider the wind load reduction effect from the worst-case estimation. However, the directionality factor is generally associated with many other factors, such as characteristics of directional wind speed and aerodynamics. To consider the effects of wind direction, the wind direction factor is introduced for calculating the design wind load. Assuming that the wind speeds in all directions are completely correlated, this study adopts the sector-by-sector method to conduct wind directionality effects studies based on the wind speed-wind direction data of a typical region [18]. The directionality factor is given as

$$K_d = P_R / P_{N,R} \tag{10}$$

$$P_R = 0.5\rho V_R^2 (C_p(\alpha))_{\text{MAX}} \tag{11}$$

$$P_{N,R} = 0.5\rho [V_R^2(\alpha) * C_p(\alpha)]_{\text{MAX}} \tag{12}$$

where $P_R$ is the wind-induced pressure for return period *R* considering the wind direction effect, $P_{N,R}$ is the wind-induced pressure for return period *R* under the worst case among all the wind directions, $\rho$ is the air density, $C_p(\alpha)$ is the extreme pressure coefficient in $\alpha$ direction, $V_R$ is a design wind speed of the worst case for period *R*, and $V_R(\alpha)$ is the design wind speed for period *R* in $\alpha$ direction.

The wind directionality factor of the 50-year return period for 16 directions is estimated in the present study. The wind speed and the wind direction data were obtained from Shanghai Longhua Meteorological Station between 1959 and 1990. The parameters of 16 wind direction extreme value distributions conforming to the Gumbel distribution are given in Figure 15. The extreme wind speed with a return period of 50 years is assessed, and then the directionality factors of all measurement points are calculated according to Equations (10)–(12).

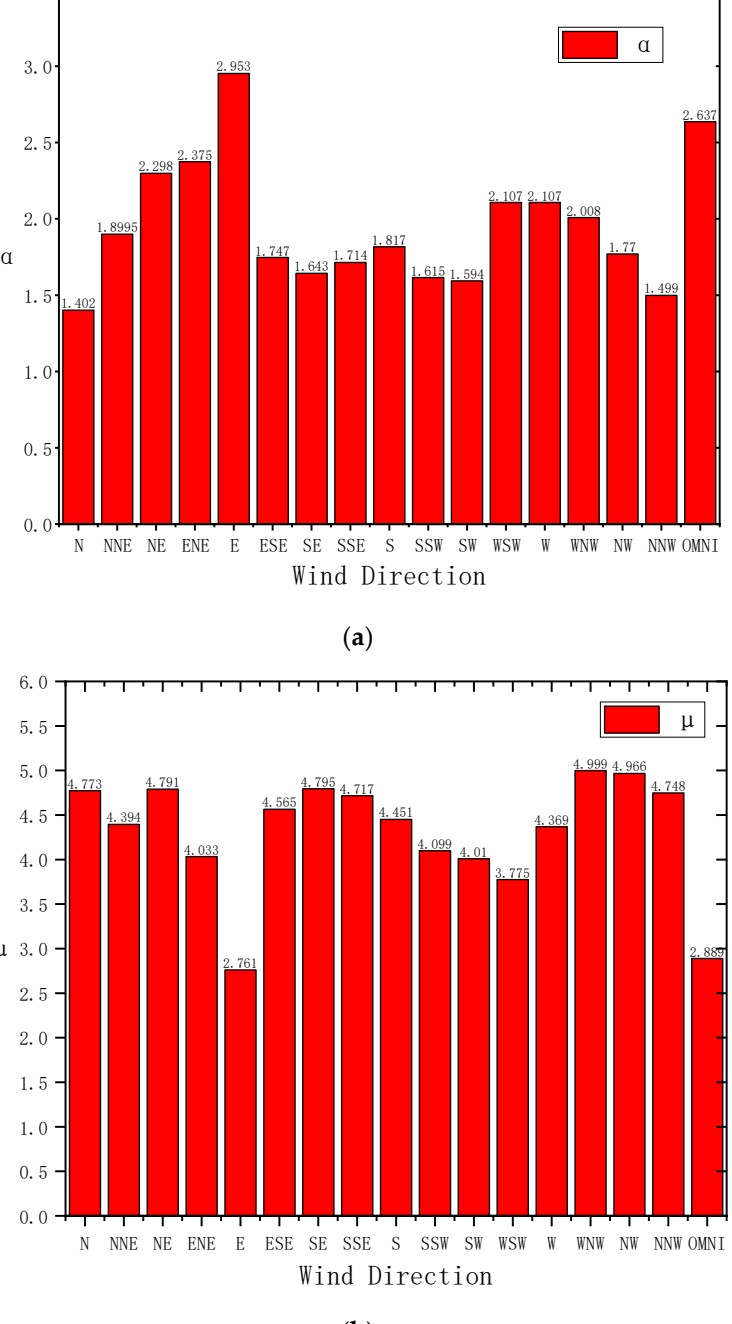

(**a**)

(**b**)

**Figure 15.** Gumbel distribution parameter estimation results of wind speed and direction joint distribution. (**a**)$\alpha$ of Gumbel distribution. (**b**) $\mu$ of Gumbel distribution.

### 3.3.1. Wind Directionality of Known Building Orientation

Selecting the windward side of the high-rise building standard model, as the north side corresponds to the wind direction N given by Yang et al. [23], it should be noted that the wind tunnel test has a wind direction interval of 10°, and the wind direction data interval is 22.5°. In this study, an interval equal to 22.5° is used to consider the wind direction effects. The corresponding wind tunnel test data are analyzed with the wind direction and the minimum wind direction angle of the building orientation.

Ten typical measuring taps on the building's facade were selected. The number of measuring points on the east elevation is 16, 19, and 22; the number of measuring points on the north elevation is 87, 106, and 110; and the number of measuring points on the west elevation is 177 and 197. The survey points on the south elevation are numbered 256 and 279. Table 3 shows the wind directionality factors of 10 typical measuring points. As can be seen from Table 3, the wind directionality factors of different measuring points are inconsistent and are lower than the standard value of 0.90.

**Table 3.** Directionality factor $K_d$ of different measuring taps for known building orientation.

| Measuring Taps Number | 16 | 19 | 22 | 87 | 106 | 110 | 177 | 197 | 256 | 279 |
|---|---|---|---|---|---|---|---|---|---|---|
| $K_d$ | 0.79 | 0.88 | 0.84 | 0.88 | 0.9 | 0.78 | 0.86 | 0.8 | 0.82 | 0.90 |

Figure 16 shows the histogram of the wind directionality factors of all the measuring taps for given building orientation. It can be observed that the wind directionality factors of each measuring tap vary within the range of 0.65–0.97, and The $K_d$ value is concentrated in the range of 0.8–0.9, with an average value of 0.85 and a coefficient of variation of 0.04.

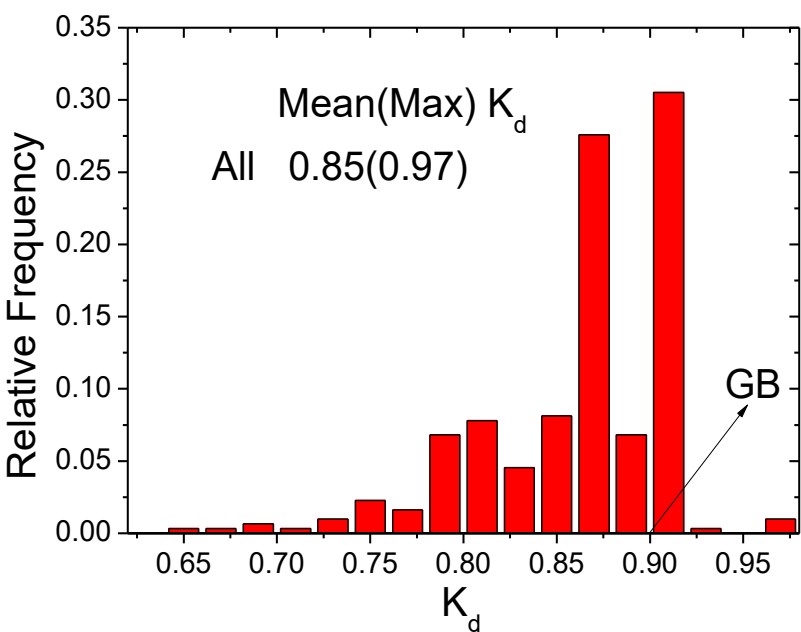

**Figure 16.** Wind directionality factors at different taps.

Figure 17 shows the probability distribution fitting results of the $K_d$ for the Gumbel and GEV probability distribution. The Gumbel distribution has a good fitting effect, and the GEV distribution has an obvious separation in some areas. Using the AIC method for testing the fitting goodness, it can be obtained that the wind directionality factor $K_d$ value is more suitable for the Gumbel distribution, and the 85% quantile value of the Gumbel distribution is about 0.90. Therefore, it can be concluded that the 0.90 wind directionality factor considered by GB50009 (2012) has a guaranteed rate of 85%.

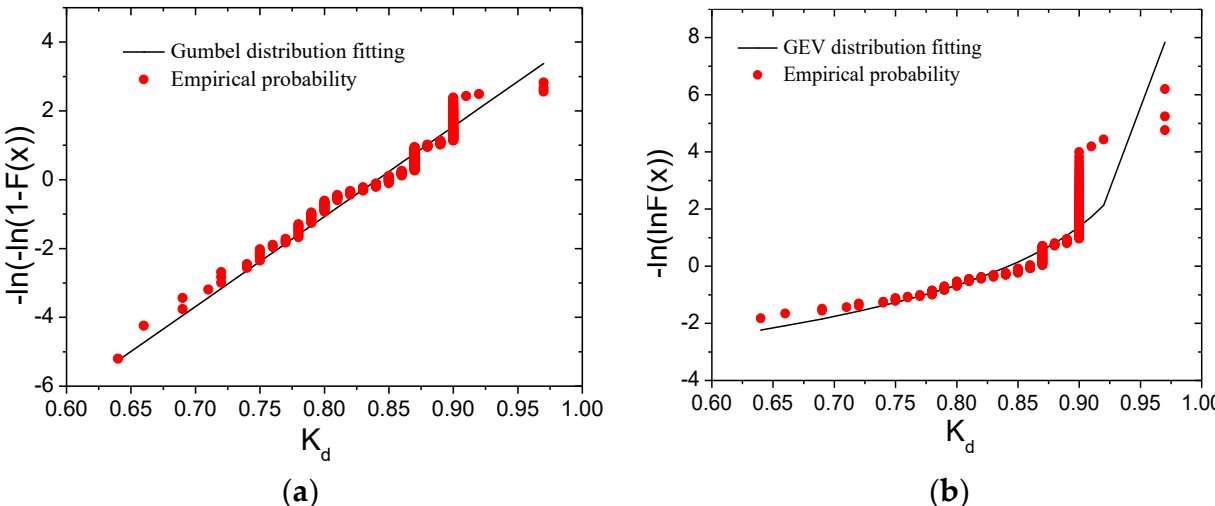

**Figure 17.** Fitting results of the probability distribution of $K_d$ value. (**a**) Gumbel probability distribution fitting, (**b**) GEV probability distribution fitting.

### 3.3.2. Wind Directionality with Unknown Building Orientation

To consider the influence of the building orientation, Laboy-Rodríguez et al. [24] proposed a method to consider the influence of the building direction on the extreme wind pressure directionality factor. Based on the known building orientation, this study considers the wind speed–wind direction relationship of Shanghai Longhua station, determining the directionality factors in Section 3.3.1. This process is repeated 16 times, and the extreme wind directionality factors based on an unknown building orientation are obtained by using Formula (13):

$$K_d = \sum_{g=1}^{16} P_r(g) K_d(g) \tag{13}$$

where $P_r(g)$ denotes the probability of occurrence in all orientations, and g represents the building orientation.

Equations (10)–(12) are used to calculate the $K_d(g)$ value of each tap for each orientation, and then the $K_d$ value of each tap considering all the building orientations is obtained by using Equation (13). Table 4 shows the $K_d(g)$ values of Tap.96 for 16 orientations. It can be found that in the table, that the wind directionality factors of the same tap in different orientations are different, and the $K_d$ value range is 0.71–0.97. Using Equation (13), the wind directionality factors are found to be 0.87 for an unknown building orientation.

**Table 4.** Directionality factors $K_d(g)$ of typical taps (Tap.96) in unknown building orientations.

| Direction (g) | 1 | 2 | 3 | 4 | 5 | 6 | 7 | 8 |
|---|---|---|---|---|---|---|---|---|
| $K_d(g)$ | 0.90 | 0.92 | 0.95 | 0.81 | 0.88 | 0.87 | 0.85 | 0.90 |
| **Direction (g)** | **9** | **10** | **11** | **12** | **13** | **14** | **15** | **16** |
| $K_d(g)$ | 0.90 | 0.83 | 0.71 | 0.89 | 0.80 | 0.80 | 0.90 | 0.97 |

Figure 18 shows the histogram and probability distribution fitting of the $K_d$ for the 308 taps with an unknown building orientation. For the normal distribution, the value of 0.88 meets the 50% guaranteed rate, and the guaranteed rate of 0.90 in Chinese code is 75%.

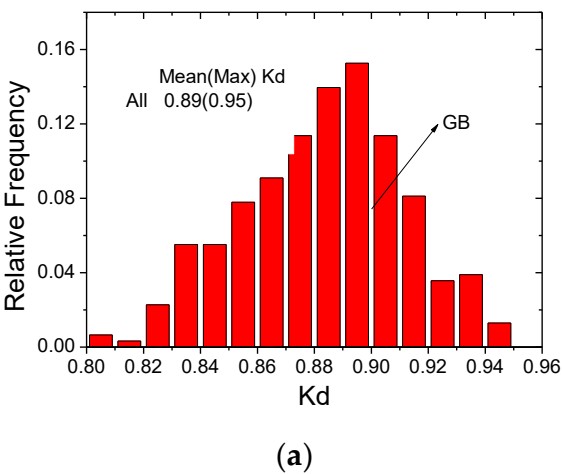
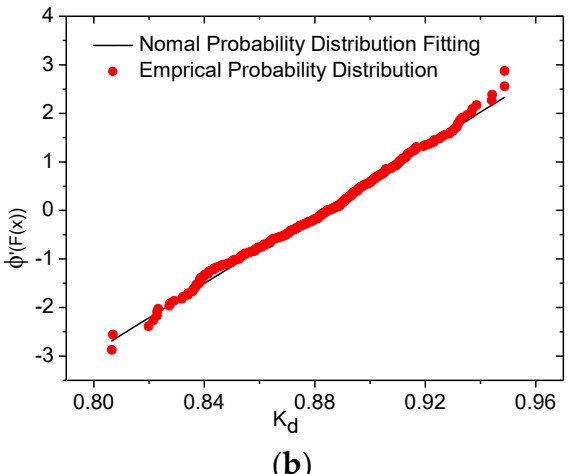

**(a)**　　　　　　　　　　　　　　　　　　　　　　**(b)**

**Figure 18.** Histogram and probability distribution fitting results of $K_d$ for unknown building orientations. (**a**) Wind directionality factors $K_d$, (**b**) Probability distribution fitting result for $K_d$.

Similarly, Table 5 shows the wind directionality factors of 10 typical taps based on the analysis with the unknown building orientation. Compared with the cases in which the building orientation is given, the $K_d$ value with the unknown building orientation is safe.

**Table 5.** Directionality factors $K_d$ (g) of typical taps in different building orientations.

| Tap Number | 16 | 19 | 22 | 87 | 106 | 110 | 177 | 197 | 256 | 279 |
|---|---|---|---|---|---|---|---|---|---|---|
| $K_d$ | 0.90 | 0.89 | 0.91 | 0.88 | 0.89 | 0.86 | 0.89 | 0.93 | 0.89 | 0.86 |

## 4. Concluding Remarks

In this study, wind tunnel tests were carried out for the CAARC building with a scale of 1/400. The probability characteristics of the extreme pressure coefficient on the building surface, the area reduction effect of the extreme pressure coefficient, and the directionality factors of the extreme wind load are especially studied. The following concluding remarks can be drawn:

(1) The probability distribution modelling is insensitive to the area size and wind direction. The GEV distribution is preferred for the probability distribution of minimum pressure coefficients.

(2) The critical extreme pressure coefficients among all the taps used in the design are conservative, and the area-averaged extreme pressure coefficient shows a better performance. Since the turbulence effect from the structure is ignored, the pressure measurements indicate that the proposed wind pressure coefficients in GB50009 (2012) may be non-conservative for the CAARC building.

(3) For the area reduction of the extreme pressure coefficient, it is observed that a larger area size would cause a smaller extreme wind load. The area reduction factors of the extreme pressure coefficients are larger than that of the local shape coefficient from the code. The area reduction factors of the extreme pressure coefficients for area sizes of 5 m$^2$, 10 m$^2$, and 25 m$^2$ vary within the range of 0.93–0.99, 0.88–0.98, and 0.85–0.97, and the COV is 0.02, 0.03, and 0.04, respectively.

(4) The mean value of the directionality factors of extreme wind load for all taps is 0.85 when the building orientation is given, and the COV is 0.04; the mean value of the extreme wind load directionality factors is 0.88 when considering the building orientation as a uniform distribution. This is close to the wind directionality factor of 0.90 given in the Chinese specification.

**Author Contributions:** Conceptualization, methodology, software, and writing, S.L. (Shouke Li) and F.X.; Investigation and validation, Y.Z., S.L. (Shouying Li) and S.Y.; Resources, supervision, and

revision, S.L. (Shouke Li), C.F. and Y.C. All authors have read and agreed to the published version of the manuscript.

**Funding:** The authors would like to gratefully acknowledge support from the National Science Foundation of China (Project No. 51508184). The work presented in this paper was also supported by grants received from the Scientific Research Fund of Hunan Provincial Education Department (19C1660).

**Institutional Review Board Statement:** Not applicable.

**Informed Consent Statement:** Not applicable.

**Data Availability Statement:** Not applicable.

**Conflicts of Interest:** The authors declare no conflict of interest.

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
