# Peer review of "Probability Characteristics, Area Reduction, and Wind Directionality Effects of Extreme Pressure Coefficients of High-Rise Buildings"

_applsci, doi:10.3390/app11157121_

Round 1

Reviewer 1 Report

General comments

The authors in this article present the results of their tests in wind tunnel for one high-rise building on a scale of 1: 400 in exposure categories D of Chinese standard GB 50009-2012, without specifying exactly what these conditions are. As a result, it is difficult for a reader outside of China to understand.

Statistical research and analyzes are generally interesting, and the subject matter is important from a utilitarian point of view. Unfortunately, when reading the article one has the compelling impression that the entire research and analysis is only useful for Chinese readers. Anyway, the research was carried out under a grant funded by the National Science Foundation of China and they were clearly focused on checking the correctness of the methods and coefficients given in the above standard. Therefore, they were used to validate its provisions. As a result, for the foreign reader, they only have cognitive value in terms of qualitative. Broader quantitative interpolations are unjustified. The question then arises: why publish an article in an international journal that is actually aimed exclusively at the Chinese reader? In order for the test to be interesting and useful for a wider group of recipients (from different countries), it should be thoroughly reedited.

The system for quoting literature should be standardized. There are three different systems in this article, for example on page 1, e.g. only reference numbers (e.g. [1-4], [12-13]); Cook and Mayne [5-6]; Chen and Huang (2010). This probably indicates that the authors used excerpts from their (or just their own?) Previous publications.

The text contains spelling and grammatical errors such as "etal" instead of "et al." etc. Requires careful editing.

The article is not suitable for publication in its current form.

Some detailed comments and suggestions are provided below.

Detailed comments

  1. Page 1. Lines # 36-38. The method propose by Peterka [5] should be a little more detailed described. Readers who do not know this method with such laconic information will not be able to understand its essence.
  2. Page 1. Lines # 40-41. What came out of this study by Chen and Huang (2010)? Without this information being clarified, it is unnecessary for the reader.
  3. Page 2. Lines # 47-48. This sentence is too laconic. These other estimation methods should be discussed in more detail or this sentence should be removed from the text.
  4. Page 2. Lines # 61-64. Authors outside China are not familiar with the requirements of the Chinese standard GB 50009-2012, so it is necessary to provide more detailed these requirements. Otherwise, this part of the text is incomprehensible for many potentially readers.
  5. Page 2. Lines # 72-82. The comments and discussion of the provisions of the Chinese standard GB 50009-2012, due to the practically lack of access to it for a wide circle of readers from outside China, are of little use and understandable. This paragraph should be removed or redrafted in a way that is useful and understandable to an international reader.
  6. Page 2. Lines # 91-92. The reference of the reader to the description of the research program to an earlier publication [21] is not correct. This program should be briefly characterized. The understanding of the article cannot be made dependent on the knowledge of the authors' previous publications.
  7. Page 2. Line # 95. What is the type of terrain exposure D? What is it characterized by?
  8. Page 3. Lines # 97-98. α = 30 and Io = 0.39 are undefined.
  9. Page 3. Lines # 107-108. The term “blockage ratio of wind tunnel tests is 0.44%” should be further explained. It is too laconic and therefore may be incomprehensible to many.
  10. Page 5. Table 1. The expression for the Gumbel distribution is not completed (probably when converting text to pdf format some part was cut).
  11. Pages 5-6. Fig.4. Quantities “μ” and “α” are not defined or explained.
  12. Page 6. Line # 176. The phrase "K-S test" is too abbreviated, especially as it is used in the text for the first time. More correct is the wording "Kolmogorov-Smirnov (K-S) test".
  13. Page 6. Line # 177. Correctly, it should be "The K-S test expression for the two attempts is as follows:"
  14. Page 7. Line # 184. The spelling "The AIC method ..." is incorrect because this abbreviation is being used for the first time. It should be “The Akaike information criterion (AIC) can also be used….”
  15. Page 10. Lines # 267-269. The EC, BEC and BG values should be more detailed described.
  16. Page 11. Lines # 278-284. The discussion concerns the provisions and possible modification of the Chinese standard. For a reader outside of China, this is unnecessary. The publications in this journal are addressed to the international community, not only to people from one country. This paragraph needs to be heavily reworded or deleted.
  17. Pages 18-19. Lines # 459-464. This conclusion is perhaps premature. Relying only one building model in the tunnel, it is unlikely to authorize the formulation of such a critical position.

Reviewer 2 Report

General comment

The manuscript “Probability characteristics, area and wind direction reduction effects of extreme pressure coefficients of high-rise buildings” has been reviewed in detail by the reviewer. The manuscript deals with a very interesting topic for the wind engineering community. The manuscript is generally well written and well structured, although some changes are required to make the document more fluent for the reader. Comments/suggestions about different sections of the manuscript are reported in detail below. The points below raised should not be considered as negative comments. 

In conclusion, the reviewer recommends a major revision before the publication.

In general, a constant mix of UK and US English has been detected throughout the whole manuscript. Please choose one and stick with that.

Title

For the reviewer is unclear the definition “wind direction reduction”. What authors mean with “reduction”? How can we reduce a wind direction? Please clarify and if necessary change the title as well as the body text accordingly.

Abstract

At line 10: please specify in terms of aerodynamics roughness length (z0) what it means.

At line 15: “may not be conservative”

At line 14-16: the sentence “The wind pressure measurements … GB50009-2012” is redundant. It might be better to rephrase here. Moreover, the acronym BEC does not seem to be an abbreviation of “area-averaged value”.

At lines 18-23: the sentence “The recommended … specification” is too long. Please split the sentence at least in three parts.

Introduction

At line 29: is a focus of what? This sentence is incomplete, please rephrase here.

At lines 40-41: the sentence ”Chen and Huang (2010) studied systematic the effects of different distribution on the extreme wind effects” is grammatically inexact (“systematic” should be “systematically”) and redundant.

At line 41: “Quan et al. (2014)”

At line 45: since the authors already introduced the acronym for “generalized extreme value”, they should use it throughout the manuscript and avoid useless repetitions.

At line 47: “The other …” which one? This is unclear for a reader.

At line 60: please mention clearly what C&C loads is, otherwise the reader would not understand.

At line 64: “lacks”

At line 70:  with the statement “considering non-hurricane wind pressures”, does it mean that this factor only applies to neutral stability conditions (synoptic winds)? Please clarify.

At line 74: what do authors mean with “full win direction”? Please clarity and rephrase eventually.

At lines 77-78: the same comment applies to “entire wind direction” and “wind direction reduction”. How a specific wind direction can be “reduced” or considered partially? Probably the authors refer to a wind sector or wind rose, please rephrase here.

At the end of the present section, please provide a short paragraph to describe the structure of the paper (as usual).

2. Experiment setup and data analysis

2.1. Wind tunnel test setup

At line 95: please specify in terms of aerodynamics roughness length (z0) what it means.

At lines 97-98: At this point of the manuscript the reader does not know what H is, whether the reference height of the building or the ABL height developed in the wind tunnel.

Fig. 2 shows a very limited (short) roughness fetch used to develop the ABL wind, but a quite long test section (from the spires to the building). Could the authors carefully describe in the body text the vortex generators and roughness fetch elements used to reproduce the ABL wind, please? And, why the test section between the spires and the wooden roughness elements is completely empty as opposite to standard procedures for ABL wind development in wind tunnel tests?

At line 110: please briefly explain why the encryption area was arranged exactly at that height.

At line 116: in the sentence “which is equivalent to one hour in full scale” please be more specific here.

2.2. Data analysis

At line 129: Could you please explain what is the reference height?

For the Eqs. 1-4, please define properly the format of the equation and keep the same margins between the equation and the corresponding number (1-4).

At line 135: what do authors mean with “tributary area”? Please clarify. Moreover, please add a space between “n” and “(n”.

At line 137: the statement “The extreme pressure coefficients of measuring taps or area average” is inexact, please rephrase here.

Please, once the acronyms (e.g. Cp) have been introduced, use them throughout the whole manuscript.

3. Results and discussion

3.1. Probability characteristics of extreme pressure coefficients

3.1.1. Extreme value distribution type

The present section is too short and does not stand alone in the present form. It could be merged to the current 3.1.2 and be deleted.

At line 146: please add a space between type “I, II, II” and the text in brackets.

Table 1: please modify as “Type I – Gumbel distribution”, “Type II - Frechet distribution”, “Type III - Weibull distribution”, and remove “extreme values or” if not needed.  

3.1.2. Probability distribution modelling of extreme pressure coefficient

At line 156: “of 2 m2, 10 m2, 25 m2 and 72 m2 (full scale)”

The authors should spend a bit more time in commenting the results of Fig. 4, at least by referring to the plotted figures and explain to the audience the meaning of k, μand α. Otherwise it does not make any sense to show all these graphs. Moreover, at line 160 please explain why the Gumbel distribution shows some deviations. For readers not fully involved in these experiments, it might not appear so obvious.

At line 176: same comment, please briefly explain what a K-S test is.

Eqs. 5-6: same comments already made for Eqs. 1-4.

At line 190-194: same comment made for (1); please explain clearly the results through observations using the graphs reported in the corresponding figures.

At line 231-233: this piece of text of conclusion should be reported in the conclusion indeed.

At line 238: please try to avoid using the first singular/plural person (“I” ,“we”) in a scientific document.

At line 242: please add a space between “2” and “m2”.

3.1.3. Comparison of extreme pressure coefficients with current specifications

At line 260: please replace “βgzμsl” with the actual “name” that explains what this terms represents.  

At line 264: could the authors explain which value of I10 has been considered in the present case? In case the authors have been considering the turbulence intensity profile of Fig.1a, it does not show any value at 10 m (i.e. 0.025 m at 1:400). The first value of turbulence value measured near the wind tunnel ground seems to be at 0.1 m (e.g. 40 m full scale).

Figures 9-10: could you please show the 4 facades and the legend (to be enlarged!) along the same row? Moreover, it would be better to re-group all pressure coefficient values in one table and do not overlap these to the building drawing.

3.2. Area reduction of extreme pressure coefficient

3.2.1. Extreme pressure coefficient for different area sizes under different wind angles

As for the previous paragraph, also in the present one the authors showed the graphs but did not make any comment/observation about the results.

At lines 277-278: the sentence “This paper … and 72 m2” is unclear for the reader and grammatically inexact. Moreover, the numbers refer to full-scale areas but the authors have not specified that.

At lines 288-289: it is difficult (or impossible) to understand where these “measuring points” are located. A figure/drawing is needed to show where these areas and measuring positions are located.

At lines 291-293: the sentence is too fragmentated and not very clear for a reader.

Figure 11 is unclear for the reader:

1) the labels (a-d) are not positioned near the corresponding image. This happens systematically to almost all figure of the manuscript. Please check this aspect carefully.

2) could authors explain why for two figures colored lines and for the other two just black and white?

3) if the x-axis shows an area (m2), why does label show “A/m2”?

3.2.2. Area reduction factor of extreme pressure coefficient

Figure 12: same comments of Fig. 11.

4. Concluding remarks

Could the authors generalize the main findings of the present study and draw a more general conclusion about the opportunity of using the present approach for similar and/or different case studies? 

Round 2

Reviewer 2 Report

The manuscript has been improved and it can be accepted for publication.